# Disambiguated Attention Embedding for Multi-Instance Partial-Label Learning

**Wei Tang**[1,2], **Weijia Zhang**[3], **Min-Ling Zhang**[1,2*]

[1] School of Computer Science and Engineering, Southeast University, Nanjing 210096, China
[2] Key Laboratory of Computer Network and Information Integration (Southeast University),
Ministry of Education, China
[3] School of Information and Physical Sciences, The University of Newcastle,
Callaghan, NSW 2308, Australia
`tangw@seu.edu.cn, weijia.zhang@newcastle.edu.au, zhangml@seu.edu.cn`

## Abstract

In many real-world tasks, the concerned objects can be represented as a multi-instance bag associated with a candidate label set, which consists of one ground-truth label and several false positive labels. Multi-instance partial-label learning (MIPL) is a learning paradigm to deal with such tasks and has achieved favorable performances. Existing MIPL approach follows the instance-space paradigm by assigning augmented candidate label sets of bags to each instance and aggregating bag-level labels from instance-level labels. However, this scheme may be suboptimal as global bag-level information is ignored and the predicted labels of bags are sensitive to predictions of negative instances. In this paper, we study an alternative scheme where a multi-instance bag is embedded into a single vector representation. Accordingly, an intuitive algorithm named DEMIPL, i.e., *Disambiguated attention Embedding for Multi-Instance Partial-Label learning*, is proposed. DEMIPL employs a disambiguation attention mechanism to aggregate a multi-instance bag into a single vector representation, followed by a momentum-based disambiguation strategy to identify the ground-truth label from the candidate label set. Furthermore, we introduce a real-world MIPL dataset for colorectal cancer classification. Experimental results on benchmark and real-world datasets validate the superiority of DEMIPL against the compared MIPL and partial-label learning approaches.

## 1 Introduction

Significant advancements in supervised machine learning algorithms have been achieved by utilizing large amounts of labeled training data. However, in numerous tasks, training data is weakly-supervised due to the substantial costs associated with data labeling [1–6]. Weak supervision can be broadly categorized into three types: incomplete, inexact, and inaccurate supervision [7]. Multi-instance learning (MIL) and partial-label learning (PLL) are typical weakly-supervised learning frameworks based on inexact supervision. In MIL, samples are represented by collections of features called bags, where each bag contains multiple instances [8–19]. Only bag-level labels are accessible during training, while instance-level labels are unknown. Consequently, the instance space in MIL contains inexact supervision, signifying that the number and location of positive instances within a positive bag remain undetermined. PLL associates each instance with a candidate label set that contains one ground-truth label and several false positive labels [20–35]. As a result, the label space in PLL embodies inexact supervision, indicating that the ground-truth label of an instance is uncertain.

---

[*]Corresponding author

37th Conference on Neural Information Processing Systems (NeurIPS 2023).

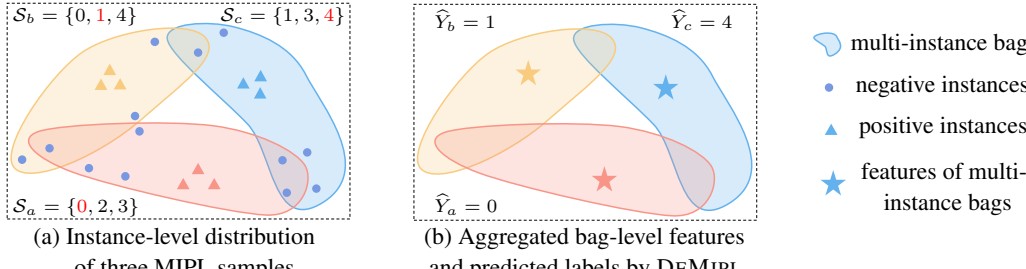

(a) Instance-level distribution of three MIPL samples

(b) Aggregated bag-level features and predicted labels by DEMIPL

Figure 1: A brief illustration of DEMIPL, where $\mathcal{S}$ and $\widehat{Y}$ are candidate label sets and predicted labels, respectively. The ground-truth labels are shown in red.

However, inexact supervision can exist in instance and label space simultaneously, i.e., dual inexact supervision [36]. This phenomenon can be observed in histopathological image classification, where an image is typically partitioned into a multi-instance bag [37–40], and labeling ground-truth labels incurs high costs due to the need for specialized expertise. Consequently, utilizing crowd-sourced candidate label sets will significantly reduce the labeling cost [41]. For this purpose, a learning paradigm called multi-instance partial-label learning (MIPL) has been proposed to work with dual inexact supervision. In MIPL, a training sample is represented as a multi-instance bag associated with a bag-level candidate label set, which comprises a ground-truth label along with several false positive labels. It is noteworthy that the multi-instance bag includes at least one instance that is affiliated with the ground-truth label, while none of the instances belong to any of the false positive labels.

Due to the difficulty in handling dual inexact supervision, to the best of our knowledge, MIPLGP [36] is the only viable MIPL approach. MIPLGP learns from MIPL data at the instance-level by utilizing a label augmentation strategy to assign an augmented candidate label set to each instance, and integrating a Dirichlet disambiguation strategy with the Gaussian processes regression model [42]. Consequently, the learned features of MIPLGP primarily capture local instance-level information, neglecting global bag-level information. This characteristic renders MIPLGP susceptible to negative instance predictions when aggregating bag-level labels from instance-level labels. As illustrated in Figure 1(a), identical or similar negative instances can simultaneously occur in multiple multi-instance bags with diverse candidate label sets, thereby intensifying the challenge of disambiguation.

In this paper, we overcome the limitations of MIPLGP by introducing a novel algorithm, named DEMIPL, i.e., *Disambiguated attention Embedding for Multi-Instance Partial-Label learning*, based on the embedded-space paradigm. Figure 1(b) illustrates that DEMIPL aggregates each multi-instance bag into a single vector representation, encompassing all instance-level features within the bag. Furthermore, DEMIPL effectively identifies the ground-truth label from the candidate label set.

Our contributions can be summarized as follows: First, we propose a disambiguation attention mechanism for learning attention scores in multi-instance bags. This is in contrast to existing attention-based MIL approaches that are limited to handling classifications with exact bag-level labels [13, 14, 43]. Second, we propose an attention loss function that encourages the attention scores of positive instances to approach one, and those of negative instances to approach zero, ensuring consistency between attention scores and unknown instance-level labels. Third, we leverage the multi-class attention scores to map the multi-instance bags into an embedded space, and propose a momentum-based disambiguation strategy to identify the ground-truth labels of the multi-instance bags from the candidate label sets. In addition, we introduce a real-world MIPL dataset for colorectal cancer classification comprising 7000 images distributed across seven categories. The candidate labels of this dataset are provided by trained crowdsourcing workers.

Experiments are conducted on the benchmark as well as real-world datasets. The experimental results demonstrate that: (a) DEMIPL achieves higher classification accuracy on both benchmark and real-world datasets. (b) The attention loss effectively enhances the disambiguation attention mechanism, accurately discerning the significance of positive and negative instances. (c) The momentum-based disambiguation strategy successfully identifies the ground-truth labels from candidate label sets, especially in scenarios with an increasing number of false positive labels.

The remainder of the paper is structured as follows. First, we introduce DEMIPL in Section 2 and present the experimental results in Section 3. Finally, we conclude the paper in Section 4.

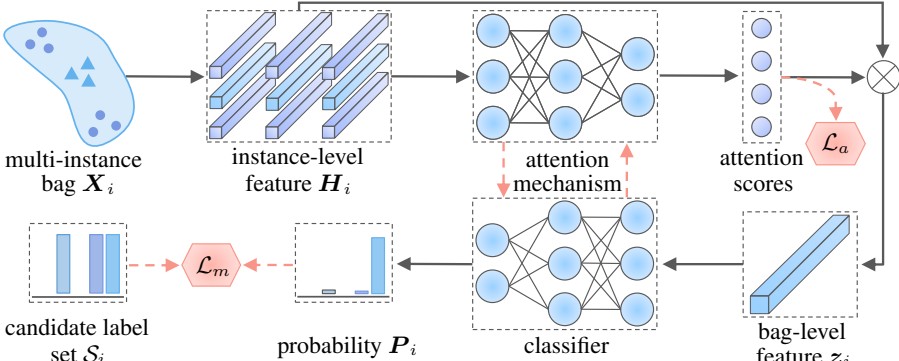

Figure 2: The framework of DEMIPL, where $\mathcal{L}_a$ and $\mathcal{L}_m$ are the attention loss and momentum-based disambiguation loss, respectively.

## 2 Methodology

### 2.1 Notations and Framework of DEMIPL

Let $\mathcal{X} = \mathbb{R}^d$ represent the instance space, and let $\mathcal{Y} = \{l_1, l_2, \cdots, l_k\}$ represent the label space containing $k$ class labels. The objective of MIPL is to derive a classifier $f : 2^{\mathcal{X}} \to \mathcal{Y}$. $\mathcal{D} = \{(\boldsymbol{X}_i, \mathcal{S}_i) \mid 1 \le i \le m\}$ is a training dataset that consists of $m$ bags and their associated candidate label sets. Particularly, $(\boldsymbol{X}_i, \mathcal{S}_i)$ is the $i$-th multi-instance partial-label sample, where $\boldsymbol{X}_i = \{\boldsymbol{x}_{i,1}, \boldsymbol{x}_{i,2}, \cdots, \boldsymbol{x}_{i,n_i}\}$ constitutes a bag with $n_i$ instances, and each instance $\boldsymbol{x}_{i,j} \in \mathcal{X}$ for $\forall j \in \{1, 2, \cdots, n_i\}$. $\mathcal{S}_i \subseteq \mathcal{Y}$ is the candidate label set that conceals the ground-truth label $Y_i$, i.e., $Y_i \in \mathcal{S}_i$. It is worth noting that the ground-truth label is unknown during the training process. Assume the latent instance-level labels within $\boldsymbol{X}_i$ is $\boldsymbol{y}_i = \{y_{i,1}, y_{i,2}, \cdots, y_{i,n_i}\}$, then $\exists y_{i,j} = Y_i$ and $\forall y_{i,j} \notin \mathcal{Y} \setminus \{Y_i\}$ hold. In the context of MIPL, an instance is considered a positive instance if its label is identical to the ground-truth label of the bag; otherwise, it is deemed a negative instance. Moreover, the class labels of negative instances do not belong to the label space.

The framework of the proposed DEMIPL is illustrated in Figure 2. It consists of three main steps. First, we extract instances in the multi-instance bag $\boldsymbol{X}_i$ and obtain instance-level feature $\boldsymbol{H}_i$. Next, we employ the disambiguation attention mechanism to integrate the multi-instance bag into a single feature vector $\boldsymbol{z}_i$. Finally, we use a classifier to predict the classification confidences $\boldsymbol{P}_i$ of the multi-instance bag. To enhance classification performance, we introduce two loss functions for model training: the attention loss $\mathcal{L}_a$ and the momentum-based disambiguation loss $\mathcal{L}_m$. During the training process, the attention mechanism and the classifier work collaboratively.

### 2.2 Disambiguation Attention Mechanism

Based on the embedded-space paradigm, a key component of DEMIPL is the disambiguation attention mechanism. The attention mechanisms are common models [44–46], which can calculate attention scores to determine the contribution of each instance to the multi-instance bag [13, 14]. The attention scores are then utilized to aggregate the instance-level features into a single vector representation.

For a multi-instance bag $\boldsymbol{X}_i = \{\boldsymbol{x}_{i,1}, \boldsymbol{x}_{i,2}, \cdots, \boldsymbol{x}_{i,n_i}\}$, we employ a neural network-based function parameterized by $h$ to extract its feature information:

$$\boldsymbol{H}_i = h(\boldsymbol{X}_i) = \{\boldsymbol{h}_{i,1}, \boldsymbol{h}_{i,2}, \cdots, \boldsymbol{h}_{i,n_i}\}, \tag{1}$$

where $\boldsymbol{h}_{i,j} = h(\boldsymbol{x}_{i,j}) \in \mathbb{R}^{d'}$ is the feature of the $j$-th instance within $i$-th bag. For the MIPL problems, we propose a multi-class attention mechanism. First, we calculate the relevance of each instance to all classes, and then transform the relevance into the contribution of each instance to the bag-level feature by a learnable linear model. The attention score $a_{i,j}$ of $\boldsymbol{x}_{i,j}$ is calculated as follows:

$$a_{i,j} = \frac{1}{1 + \exp\left\{-\boldsymbol{W}^{\top}\left(\tanh\left(\boldsymbol{W}_v^{\top}\boldsymbol{h}_{i,j} + \boldsymbol{b}_v\right) \odot \mathrm{sigm}\left(\boldsymbol{W}_u^{\top}\boldsymbol{h}_{i,j} + \boldsymbol{b}_u\right)\right)\right\}}, \tag{2}$$

where $\boldsymbol{W}^\top \in \mathbb{R}^{1 \times k}$, $\boldsymbol{W}_v^\top$, $\boldsymbol{W}_u^\top \in \mathbb{R}^{k \times d'}$, and $\boldsymbol{b}_v$, $\boldsymbol{b}_u \in \mathbb{R}^k$ are parameters of the attention mechanism. $\tanh(\cdot)$ and $\text{sigm}(\cdot)$ are the hyperbolic tangent and sigmoid functions to generate non-linear outputs for the models, respectively. $\odot$ represents an element-wise multiplication. Consequently, the bag-level feature is aggregated by weighted sums of instance-level features:

$$\boldsymbol{z}_i = \frac{1}{\sum_{j=1}^{n_i} a_{i,j}} \sum_{j=1}^{n_i} a_{i,j} \boldsymbol{h}_{i,j}, \tag{3}$$

where $\boldsymbol{z}_i$ is the bag-level feature of $\boldsymbol{X}_i$. To ensure that the aggregated features accurately represent the multi-instance bag, it is necessary to maintain the consistency between attention scores and instance-level labels, that is, the attention scores of positive instances should be significantly higher than those of negative instances. To achieve this, the proposed attention loss is shown below:

$$\mathcal{L}_a = -\frac{1}{m} \sum_{i=1}^m \sum_{j=1}^{n_i} a_{i,j} \log a_{i,j}. \tag{4}$$

Different from existing attention-based MIL approaches where most of them can only handle binary classification, DEMIPL produces multi-class attention scores using Equation (2). Furthermore, unlike loss-based attention [14] that extends binary attention score to multi-class using a naive softmax function, DEMIPL utilizes a learnable model with the attention loss to encourage attention scores of negative and positive instances to approach $0$ and $1$, respectively. The ambiguity in attention scores is reduced since the differences in attention scores between positive and negative instances are amplified. As a result, the disambiguated attention scores can make bag-level vector representations discriminative, thereby enabling the classifier to accurately identify ground-truth labels.

## 2.3 Momentum-based Disambiguation Strategy

After obtaining the bag-level feature, the goal is to accurately identify the ground-truth label from the candidate label set. Therefore, we propose a novel disambiguation strategy, namely using the momentum-based disambiguation loss to compute the weighted sum of losses for each category. Specifically, the proposed momentum-based disambiguation loss is defined as follows:

$$\mathcal{L}_m = \frac{1}{m} \sum_{i=1}^m \sum_{c=1}^k w_{i,c}^{(t)} \ell\left(f_c^{(t)}(\boldsymbol{z}_i^{(t)}), \mathcal{S}_i\right), \tag{5}$$

where $(t)$ refers to the $t$-th epoch. $\boldsymbol{z}_i^{(t)}$ is the bag-level feature of multi-instance bag $\boldsymbol{X}_i$ and $f_c^{(t)}(\cdot)$ is the model output on the $c$-th class at the $t$-th epoch. $\ell(\cdot)$ is the cross-entropy loss, and $w_{i,c}^{(t)}$ weights the loss value on the $c$-th class at the $t$-th epoch.

Following the principle of the identification-based disambiguation strategy [47], the label with the minimal loss value on the candidate label set can be considered the ground-truth label. We aim to assign a weight of $1$ to the single ground-truth label and a weight of $0$ to the rest of the candidate labels. However, the ground-truth label is unknown during the training process. To overcome this issue, we allocate weights based on the magnitude of class probabilities, ensuring that larger class probabilities are associated with higher weights. Specifically, we initialize the weights by:

$$w_{i,c}^{(0)} = \begin{cases} \frac{1}{|\mathcal{S}_i|} & \text{if } Y_{i,c} \in \mathcal{S}_i, \\ 0 & \text{otherwise}, \end{cases} \tag{6}$$

where $\frac{1}{|\mathcal{S}_i|}$ is the cardinality of the candidate label set $\mathcal{S}_i$. The weights are updated as follows:

$$w_{i,c}^{(t)} = \begin{cases} \lambda^{(t)} w_{i,c}^{(t-1)} + (1 - \lambda^{(t)}) \frac{f_c^{(t)}(\boldsymbol{z}_i^{(t)})}{\sum_{j \in \boldsymbol{s}_i} f_j^{(t)}(\boldsymbol{z}_j^{(t)})} & \text{if } Y_{i,c} \in \mathcal{S}_i, \\ 0 & \text{otherwise}, \end{cases} \tag{7}$$

where the momentum parameter $\lambda^{(t)} = \frac{T-t}{T}$ is a trade-off between the weights at the last epoch and the outputs at the current epoch. $T$ is the maximum training epoch.

It is worth noting that the momentum-based disambiguation strategy is a general form of the progressive disambiguation strategy. Specifically, when $\lambda^{(t)} = 0$, the momentum-based disambiguation strategy degenerates into the progressive disambiguation strategy [47]. When $\lambda^{(t)} = 1$, the momentum-based disambiguation strategy degenerates into the averaging-based disambiguation strategy [21], which equally treats every candidate label.

## 2.4 Synergy between Attention Mechanism and Disambiguation Strategy

Combining the attention loss and disambiguation loss, the full loss function is derived as follows:

$$\mathcal{L} = \mathcal{L}_m + \lambda_a \mathcal{L}_a, \tag{8}$$

where $\lambda_a$ serves as a constant weight for the attention loss. In each iteration, the disambiguation attention mechanism aggregates a discriminative vector representation for each multi-instance bag. Subsequently, the momentum-based disambiguation strategy takes that feature as input and yields the disambiguated candidate label set, i.e., class probabilities. Meanwhile, the attention mechanism relies on the disambiguated candidate label set to derive attention scores. Thus, the disambiguation attention mechanism and the momentum-based disambiguation strategy work collaboratively.

## 3 Experiments

### 3.1 Experimental Setup

**Benchmark Datasets** We utilize four benchmark MIPL datasets stemming from MIPLGP literature [36], i.e., MNIST-MIPL, FMNIST-MIPL, Birdsong-MIPL, and SIVAL-MIPL from domains of image and biology [48–51]. Table 1 summarizes the characteristics of both the benchmark and real-world datasets. We use *#bags*, *#ins*, *#dim*, *avg. #ins*, *#class*, and *avg. #CLs* to denote the number of bags, number of instances, dimension of each instance, average number of instances in all bags, number of class labels, and the average size of candidate label set in each dataset.

**Real-World Dataset** We introduce CRC-MIPL, the first real-world MIPL dataset for colorectal cancer classification (CRC). It comprises 7000 hematoxylin and eosin (H&E) staining images taken from colorectal cancer and normal tissues. Each image has dimensions of $224 \times 224$ pixels and is categorized into one of the seven classes based on the tissue cell types. CRC-MIPL is derived from a larger dataset used for colorectal cancer classification, which originally contains 100000 images with nine classes [52]. The adipose and background classes exhibit significant dissimilarities compared to the other categories. Therefore, we choose the remaining seven classes to sample 1000 images per class. These classes include debris, lymphocytes, mucus, smooth muscle, normal colon mucosa, cancer-associated stroma, and colorectal adenocarcinoma epithelium.

We employ four image bag generators [53]: Row [54], single blob with neighbors (SBN) [54], k-means segmentation (KMeansSeg) [55], and scale-invariant feature transform (SIFT) [56], to obtain a bag of instances from each image, respectively. The candidate label sets of CRC-MIPL are provided by three crowdsourcing workers without expert pathologists. Each of the workers annotates all 7000 images, and each worker assigned candidate labels with non-zero probabilities to form a label set per image. A higher probability indicates a higher likelihood of being the ground-truth label, while a probability of zero implies the label is a non-candidate label. After obtaining three label sets for each image, we distill a final candidate label set as follows. A label present in two or three label sets is selected as a member of the final candidate label set. If the final candidate label set consists of only one or no label, we pick the labels corresponding to the highest probability in each label set. The average length of the final candidate label set per image is 2.08. More detailed information on the MIPL datasets can be found in the Appendix.

**Compared Algorithms** For comparative studies, we consider one MIPL algorithm MIPLGP [36] and four PLL algorithms, containing one feature-aware disambiguation algorithm PL-AGGD [32] and three deep learning-based algorithms, namely PRODEN [47], RC [57], and LWS [58].

Table 1: Characteristics of the Benchmark and Real-World MIPL Datasets.

| Dataset | #bags | #ins | #dim | avg. #ins | #class | avg. #CLs | domain |
|---------|-------|------|------|-----------|--------|-----------|--------|
| MNIST-MIPL | 500 | 20664 | 784 | 41.33 | 5 | 2, 3, 4 | image |
| FMNIST-MIPL | 500 | 20810 | 784 | 41.62 | 5 | 2, 3, 4 | image |
| Birdsong-MIPL | 1300 | 48425 | 38 | 37.25 | 13 | 2, 3, 4, 5, 6, 7 | biology |
| SIVAL-MIPL | 1500 | 47414 | 30 | 31.61 | 25 | 2, 3, 4 | image |
| CRC-MIPL-Row | 7000 | 56000 | 9 | 8 | 7 | 2.08 | image |
| CRC-MIPL-SBN | 7000 | 63000 | 15 | 9 | 7 | 2.08 | image |
| CRC-MIPL-KMeansSeg | 7000 | 30178 | 6 | 4.311 | 7 | 2.08 | image |
| CRC-MIPL-SIFT | 7000 | 175000 | 128 | 25 | 7 | 2.08 | image |

**Implementation** DEMIPL is implemented using PyTorch [59] on a single Nvidia Tesla V100 GPU. We employ the stochastic gradient descent (SGD) optimizer with a momentum of $0.9$ and weight decay of $0.0001$. The initial learning rate is chosen from a set of $\{0.01, 0.05\}$ and is decayed using a cosine annealing method [60]. The number of epochs is set to $200$ for the SIVAL-MIPL and CRC-MIPL datasets, and $100$ for the remaining three datasets. The value of $\lambda_a$ is selected from a set of $\{0.0001, 0.001\}$. For the MNIST-MIPL and FMNIST-MIPL datasets, we utilize a two-layer CNN in work of Ilse et al. [13] as a feature extraction network, whereas for the remaining datasets, no feature extraction network is employed. To ensure the reliability of the results, we conduct ten runs of random train/test splits with a ratio of $7 : 3$ for all datasets. The mean accuracies and standard deviations are recorded for each algorithm. Subsequently, we perform pairwise t-test at a significance level of $0.05$.

To map MIPL data into PLL data, we employ the Mean strategy and the MaxMin strategy as described in the MIPLGP literature [36]. The Mean strategy calculates the average values of each feature dimension for all instances within a multi-instance bag to yield a bag-level vector representation. The MaxMin strategy involves computing the maximum and minimum values of each feature dimension for all instances within a multi-instance bag and concatenating them to construct bag-level features. In the subsequent section, we report the results of PRODEN, RC, and LWS using linear models, while the results with multi-layer perceptrons are provided in the Appendix.

## 3.2 Experimental Results on the Benchmark Datasets

Table 2 presents the classification results achieved by DEMIPL and the compared algorithms on the benchmark datasets with varying numbers of false positive labels $r$. Compared to MIPLGP, DEMIPL demonstrates superior performance in $8$ out of $12$ cases, with no significant difference observed in the remaining $1$ out of $12$ cases. MIPLGP performs better than DEMIPL on the SIVAL-MIPL dataset, primarily due to the unique characteristics of the SIVAL-MIPL dataset. The dataset encompasses $25$ highly diverse categories, such as apples, medals, books, and shoes, resulting in distinctive and discriminative features within multi-instance bags. Each instance's feature includes color and texture information derived from the instance itself as well as its four cardinal neighbors, which enhances the distinctiveness of instance-level features. This suggests that instances with similar features are rarely found across different multi-instance bags. Therefore, by following the instance-space paradigm, MIPLGP effectively leverages these distinctive attributes of the dataset.

Compared to PLL algorithms, DEMIPL outperforms them on all benchmark datasets. This superiority can be attributed to two main factors. First, PLL algorithms cannot directly handle multi-instance bags, whereas the original multi-instance features possess better discriminative power than the degenerated features obtained through the Mean and MaxMin strategies. Second, the proposed momentum-based disambiguation strategy is more robust than the disambiguation strategies of the compared algorithms. It should be noted that although these PLL algorithms achieve satisfactory results in PLL tasks, their performance in addressing MIPL problems is inferior to dedicated MIPL algorithms, namely DEMIPL and MIPLGP. This observation emphasizes the greater challenges posed by MIPL problems, which involve increased ambiguity in supervision compared to PLL problems, and highlights the necessity of developing specialized algorithms for MIPL.

Additionally, we experiment with another extension of applying PLL algorithms to MIPL data by directly assigning a bag-level candidate label set as the candidate label set for each instance within the bag. However, all of them perform worse than MIPL. Moreover, the majority of the compared PLL algorithms fail to produce satisfactory results. This is likely caused by the fact that the ground-truth labels for most instances are absent from their respective candidate label sets. Consequently, the absences of ground-truth labels impede the disambiguation ability of MIPL algorithms.

## 3.3 Experimental Results on the Real-World Dataset

The classification accuracy of DEMIPL and the compared algorithms on the CRC-MIPL dataset is presented in Table 3, where the symbol – indicates that MIPLGP encounters memory overflow issues on our V100 GPUs. DEMIPL demonstrates superior performance compared to MIPLGP on the CRC-MIPL-SBN and CRC-MIPL-KMeansSeg datasets, while only falling behind MIPLGP on the CRC-MIPL-Row dataset. When compared to the PLL algorithms, DEMIPL achieves better results in $28$ out of $32$ cases, and only underperforms against PL-AGGD in $2$ cases on CRC-MIPL-Row and CRC-MIPL-SBN.

Table 2: Classification accuracy (mean±std) of each comparing algorithm on the benchmark datasets in terms of the different number of false positive labels [$r \in \{1, 2, 3\}$]. ●/○ indicates whether the performance of DEMIPL is statistically superior/inferior to the compared algorithm on each dataset.

| Algorithm | $r$ | MNIST-MIPL | FMNIST-MIPL | Birdsong-MIPL | SIVAL-MIPL |
|---|---|---|---|---|---|
| DEMIPL | 1 | 0.976±0.008 | 0.881±0.021 | 0.744±0.016 | 0.635±0.041 |
| | 2 | 0.943±0.027 | 0.823±0.028 | 0.701±0.024 | 0.554±0.051 |
| | 3 | 0.709±0.088 | 0.657±0.025 | 0.696±0.024 | 0.503±0.018 |
| MIPLGP | 1 | 0.949±0.016● | 0.847±0.030● | 0.716±0.026● | 0.669±0.019○ |
| | 2 | 0.817±0.030● | 0.791±0.027● | 0.672±0.015● | 0.613±0.026○ |
| | 3 | 0.621±0.064● | 0.670±0.052 | 0.625±0.015● | 0.569±0.032○ |
| Mean | | | | | |
| PRODEN | 1 | 0.605±0.023● | 0.697±0.042● | 0.296±0.014● | 0.219±0.014● |
| | 2 | 0.481±0.036● | 0.573±0.026● | 0.272±0.019● | 0.184±0.014● |
| | 3 | 0.283±0.028● | 0.345±0.027● | 0.211±0.013● | 0.166±0.017● |
| RC | 1 | 0.658±0.031● | 0.753±0.042● | 0.362±0.015● | 0.279±0.011● |
| | 2 | 0.598±0.033● | 0.649±0.028● | 0.335±0.011● | 0.258±0.017● |
| | 3 | 0.392±0.033● | 0.401±0.063● | 0.298±0.009● | 0.237±0.020● |
| LWS | 1 | 0.463±0.048● | 0.726±0.031● | 0.265±0.010● | 0.240±0.014● |
| | 2 | 0.209±0.028● | 0.720±0.025● | 0.254±0.010● | 0.223±0.008● |
| | 3 | 0.205±0.013● | 0.579±0.041● | 0.237±0.005● | 0.194±0.026● |
| PL-AGGD | 1 | 0.671±0.027● | 0.743±0.026● | 0.353±0.019● | 0.355±0.015● |
| | 2 | 0.595±0.036● | 0.677±0.028● | 0.314±0.018● | 0.315±0.019● |
| | 3 | 0.380±0.032● | 0.474±0.057● | 0.296±0.015● | 0.286±0.018● |
| MaxMin | | | | | |
| PRODEN | 1 | 0.508±0.024● | 0.424±0.045● | 0.387±0.014● | 0.316±0.019● |
| | 2 | 0.400±0.037● | 0.377±0.040● | 0.357±0.012● | 0.287±0.024● |
| | 3 | 0.345±0.048● | 0.309±0.058● | 0.336±0.012● | 0.250±0.018● |
| RC | 1 | 0.519±0.028● | 0.731±0.027● | 0.390±0.014● | 0.306±0.023● |
| | 2 | 0.469±0.035● | 0.666±0.027● | 0.371±0.013● | 0.288±0.021● |
| | 3 | 0.380±0.048● | 0.524±0.034● | 0.363±0.010● | 0.267±0.020● |
| LWS | 1 | 0.242±0.042● | 0.435±0.049● | 0.225±0.038● | 0.289±0.017● |
| | 2 | 0.239±0.048● | 0.406±0.040● | 0.207±0.034● | 0.271±0.014● |
| | 3 | 0.218±0.017● | 0.318±0.064● | 0.216±0.029● | 0.244±0.023● |
| PL-AGGD | 1 | 0.527±0.035● | 0.391±0.040● | 0.383±0.014● | 0.397±0.028● |
| | 2 | 0.439±0.020● | 0.371±0.037● | 0.372±0.020● | 0.360±0.029● |
| | 3 | 0.321±0.043● | 0.327±0.028● | 0.344±0.011● | 0.328±0.023● |

Table 3: Classification accuracy (mean±std) of each comparing algorithm on the CRC-MIPL dataset.

| Algorithm | CRC-MIPL-Row | CRC-MIPL-SBN | CRC-MIPL-KMeansSeg | CRC-MIPL-SIFT |
|---|---|---|---|---|
| DEMIPL | 0.408±0.010 | 0.486±0.014 | 0.521±0.012 | 0.532±0.013 |
| MIPLGP | 0.432±0.005○ | 0.335±0.006● | 0.329±0.012● | – |
| Mean | | | | |
| PRODEN | 0.365±0.009● | 0.392±0.008● | 0.233±0.018● | 0.334±0.029● |
| RC | 0.214±0.011● | 0.242±0.012● | 0.226±0.009● | 0.209±0.007● |
| LWS | 0.291±0.010● | 0.310±0.006● | 0.237±0.008● | 0.270±0.007● |
| PL-AGGD | 0.412±0.008 | 0.480±0.005● | 0.358±0.008● | 0.363±0.012● |
| MaxMin | | | | |
| PRODEN | 0.401±0.007 | 0.447±0.011● | 0.265±0.027● | 0.291±0.011● |
| RC | 0.227±0.012● | 0.338±0.010● | 0.208±0.007● | 0.246±0.008● |
| LWS | 0.299±0.008● | 0.382±0.009● | 0.247±0.005● | 0.230±0.007● |
| PL-AGGD | 0.460±0.008○ | 0.524±0.008○ | 0.434±0.009● | 0.285±0.009● |

The above results indicate that DEMIPL achieves the best performance when combined with stronger bag generators such as CRC-MIPL-KMeansSeg and CRC-MIPL-SIFT. This combination enables the disambiguation attention mechanism to learn meaningful embeddings. This aligns with the fact that CRC-MIPL-Row and CRC-MIPL-SBN use classic multi-instance bag generators that only consider pixel colors, and lack the ability to extract any of the content information. In contrast, CRC-MIPL-KMeansSeg and CRC-MIPL-SIFT are content-aware generators that are capable of producing semantically meaningful features. Both CRC-MIPL-Row and CRC-MIPL-SBN segment images using fixed grids, and represents instances based on their pixel-level colors and the colors of their adjacent rows or grids. Consequently,

Table 4: Classification accuracy (mean±std) of DEMIPL-MD and DEMIPL.

| Algorithm | FMNIST-MIPL | | | SIVAL-MIPL | | |
|---|---|---|---|---|---|---|
| | $r = 1$ | $r = 2$ | $r = 3$ | $r = 1$ | $r = 2$ | $r = 3$ |
| DEMIPL-MD | 0.744±0.273 | 0.784±0.018 | 0.586±0.101 | 0.607±0.024 | 0.530±0.021 | 0.499±0.035 |
| DEMIPL | 0.881±0.021 | 0.823±0.028 | 0.657±0.025 | 0.635±0.041 | 0.554±0.051 | 0.503±0.018 |

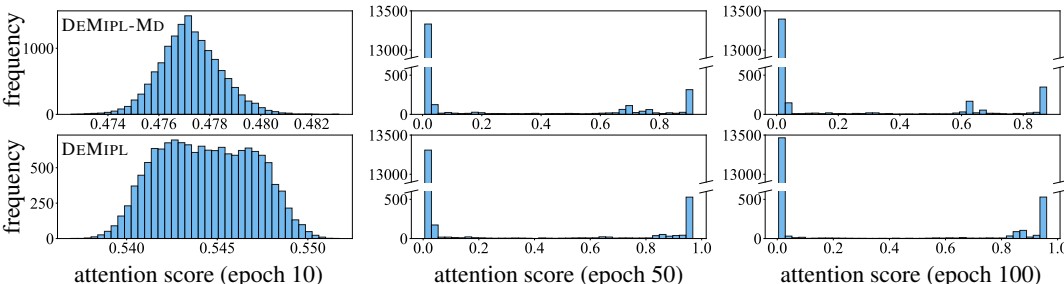

Figure 3: The frequency distribution of attention scores on MNIST-MIPL dataset ($r = 1$).

instances in CRC-MIPL-Row and CRC-MIPL-SBN exhibit similar feature representations, and possess limited discriminative power when distinguishing positive and negative instances. With more powerful bag generators such as CRC-MIPL-KMeansSeg and CRC-MIPL-SIFT, which generate content-aware features that are more informative and discriminative, the disambiguation power of DEMIPL can be fully utilized as demonstrated by the significant performance advantages against all compared baselines.

Furthermore, the CRC-MIPL dataset exhibits distinct differences between tissue cells and the background in each image. The Mean strategy diminishes the disparities and discriminations, leading to superior outcomes for the Maxmin strategy in most cases when compared to the Mean strategy.

### 3.4 Further Analysis

**Effectiveness of the Attention Loss** To validate the effectiveness of the attention loss, we introduce a degenerated variant named DEMIPL-MD, which excludes the attention loss from DEMIPL. Table 4 verifies that DEMIPL achieves superior accuracy compared to DEMIPL-MD on both the FMNIST-MIPL and SIVAL-MIPL datasets. Notably, the difference is more pronounced on the FMNIST-MIPL dataset than that on the SIVAL-MIPL dataset. This can be attributed to the fact that the feature representation of each instance in the FMNIST-MIPL dataset solely comprises self-contained information, enabling clear differentiation between positive and negative instances. Conversely, the feature representation of each instance in the SIVAL-MIPL dataset encompasses both self and neighboring information, leading to couplings between the feature information of positive instances and negative instances.

To further investigate the scores learned by the attention loss, we visualize the frequency distribution of attention scores throughout the training process. As illuminated in Figure 3, the top row corresponds to DEMIPL-MD, while the bottom row corresponds to DEMIPL. At epoch=10, attention scores generated by DEMIPL show higher dispersion, suggesting that DEMIPL trains faster than DEMIPL-MD. At epoch=50 and 100, attention scores computed by DEMIPL tend to converge towards two extremes: attention scores for negative instances gravitate towards zero, while attention scores for positive instances approach one. In conclusion, the attention loss is conducive to calculating appropriate attention scores for positive and negative instances, thereby improving accuracy.

**Effectiveness of the Momentum-based Disambiguation Strategy** To further investigate the momentum-based disambiguation strategy, the performance of DEMIPL is compared with its two degenerated versions denoted as DEMIPL-PR and DEMIPL-AV. DEMIPL-PR is obtained by setting the momentum parameter $\lambda^{(t)} = 0$ in Equation (7), which corresponds to progressively updating the weights based on the current output of the classifier. In contrast, DEMIPL-AV is obtained by setting the momentum parameter $\lambda^{(t)} = 1$, resulting in uniform weights throughout the training process.

Figure 4 illustrates the performance comparison among DEMIPL, DEMIPL-PR, and DEMIPL-AV on the MNIST-MIPL, FMNIST-MIPL, and Birdsong-MIPL datasets. When the number of false positive labels is small, DEMIPL-PR and DEMIPL-AV demonstrate similar performance to DEMIPL. However,

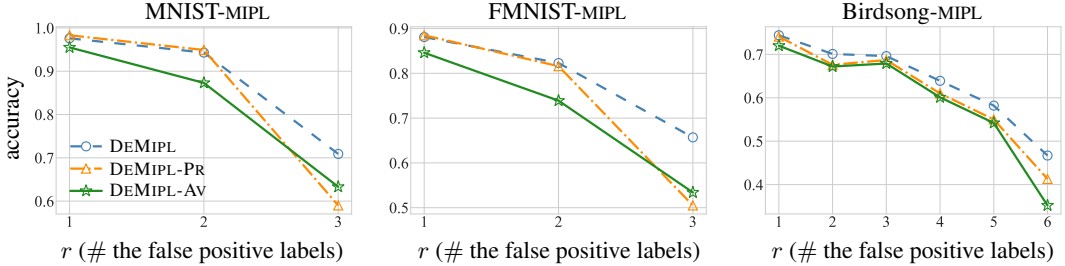

Figure 4: Classication accuracy of DEMIPL, DEMIPL-PR, and DEMIPL-AV with varying $r$.

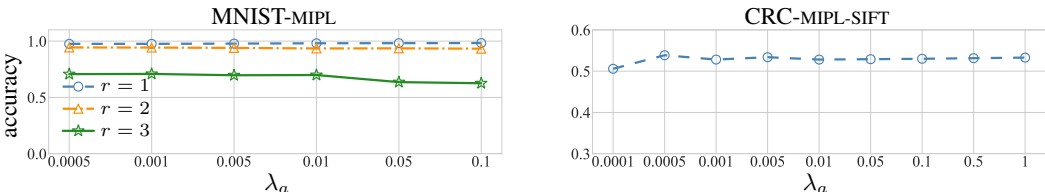

Figure 5: Performance of DEMIPL with varying weight $\lambda_a$.

as the number of false positive labels increases, DEMIPL consistently outperforms DEMIPL-PR and DEMIPL-AV by a significant margin. This observation suggests that the momentum-based disambiguation strategy is more robust in handling higher levels of disambiguation complexity. Furthermore, it can be observed that DEMIPL-PR generally outperforms DEMIPL-AV across various scenarios. However, when $r = 3$ in the MNIST-MIPL and FMNIST-MIPL datasets, DEMIPL-AV surpasses DEMIPL-PR. We believe this can be attributed to the following reason: having three false positive labels within the context of five classifications represents an extreme case. DEMIPL-PR likely assigns higher weights to false positive labels, whereas DEMIPL-AV uniformly assigns weights to each candidate label, adopting a more conservative approach to avoid assigning excessive weights to false positive labels. In a nutshell, the proposed momentum-based disambiguation strategy demonstrates superior robustness compared to existing methods for disambiguation.

**Parameter Sensitivity Analysis** The weight $\lambda_a$ in Equation (8) is the primary hyperparameter in DEMIPL. Figure 5 illustrates the sensitivity analysis of the weight $\lambda_a$ on the MNIST-MIPL and CRC-MIPL-SIFT datasets. The learning rates on the MNIST-MIPL dataset are set to 0.01, 0.01, 0.05 for $r = 1$, 2, 3, respectively, while on the CRC-MIPL-SIFT dataset, the learning rate is set to 0.01. As illuminated in Figure 5, DEMIPL demonstrates insensitivity to changes in the weight $\lambda_a$. In the experiments involving DEMIPL and its variants, the weight $\lambda_a$ is chosen from a set of $\{0.0001, 0.001\}$.

## 4 Conclusion

In this paper, we propose DEMIPL, the first deep learning-based algorithm for multi-instance partial-label learning, accompanied by a real-world dataset. Specifically, DEMIPL utilizes the disambiguation attention mechanism to aggregate each multi-instance bag into a single vector representation, which is further used in conjunction with the momentum-based disambiguation strategy to determine the ground-truth label from the candidate label set. The disambiguation attention mechanism and momentum-based strategy synergistically facilitate disambiguation in both the instance space and label space. Extensive experimental results indicate that DEMIPL outperforms the compared algorithms in 96.3% of cases on benchmark datasets and 85.7% of cases on the real-world dataset.

Despite DEMIPL's superior performance compared to the well-established MIPL and PLL approaches, it exhibits certain limitations and there are several unexplored research avenues. For example, DEMIPL assumes independence among instances within each bag. A promising avenue for future research involves considering dependencies between instances. Moreover, akin to MIL algorithms grounded in the embedded-space paradigm [13], accurately predicting instance-level labels poses a challenging endeavor. One possible approach entails the introduction of an instance-level classifier.

## Acknowledgements

The authors wish to thank the anonymous reviewers for their helpful comments and suggestions. This work was supported by the National Science Foundation of China (62225602, 62206047), the Postgraduate Research & Practice Innovation Program of Jiangsu Province (KYCX23_0317), and the Big Data Computing Center of Southeast University.

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

# A Appendix

## A.1 Pseudo-Code of DEMIPL

Given an unseen multi-instance bag $\boldsymbol{X}_* = [\boldsymbol{x}_{*,1}, \boldsymbol{x}_{*,2}, \cdots, \boldsymbol{x}_{*,n_*}]$ with $n_*$ instances, DEMIPL initially utilizes the feature extractor to obtain the instance-level representation as follows:

$$\boldsymbol{H}_* = h(\boldsymbol{X}_*) = \{\boldsymbol{h}_{*,1}, \boldsymbol{h}_{*,2}, \cdots, \boldsymbol{h}_{*,n_*}\}. \tag{9}$$

Subsequently, DEMIPL maps the instance-level representation $\boldsymbol{H}_*$ into a vector representation $\boldsymbol{z}_*$ using the disambiguation attention mechanism, which is described as follows:

$$a_{*,j} = \frac{1}{1 + \exp\left\{-\boldsymbol{W}^\top \left(\tanh\left(\boldsymbol{W}_v^\top \boldsymbol{h}_{*,j} + \boldsymbol{b}_v\right) \odot \mathrm{sigm}\left(\boldsymbol{W}_u^\top \boldsymbol{h}_{*,j} + \boldsymbol{b}_u\right)\right)\right\}}, \tag{10}$$

$$\boldsymbol{z}_* = \frac{1}{\sum_{j=1}^{n_*} a_{*,j}} \sum_{j=1}^{n_*} a_{*,j} \boldsymbol{h}_{*,j}. \tag{11}$$

Finally, we employ the trained classifier $\mathbf{f}$ to predict the label of $\boldsymbol{X}_*$ by:

$$Y_* = \arg\max_{c \in \mathcal{Y}} f_c(\boldsymbol{z}_*), \tag{12}$$

where $f_c(\cdot)$ is the $c$-th element of $\mathbf{f}(\cdot)$, and the normalized function $f_c(\cdot)$ represents the probability of class label $c$ being the ground-truth label.

Algorithm 1 summarizes the complete procedure of DEMIPL. First, the algorithm uniformly initializes the weights of the momentum-based disambiguation loss (Step 1). Next, the model training can be divided into two sub-steps (Steps 2-13). The initial sub-step involves extracting features for each mini-batch and aggregating them into bag-level vector representations (Steps 5-7). The subsequent sub-step encompasses calculating the loss function and updating the model (Steps 8-11). Finally, for an unseen multi-instance bag, instance-level features are extracted and aggregated into a bag-level vector representation, which is used to predict the label (Steps 14-16).

---

**Algorithm 1** $Y_* = $ DEMIPL $(\mathcal{D}, \lambda_a, T, \boldsymbol{X}_*)$

---

**Inputs**:
$\mathcal{D}$ : the multi-instance partial-label training set $\{(\boldsymbol{X}_i, \mathcal{S}_i) \mid 1 \leq i \leq m\}$, where $\boldsymbol{X}_i = \{\boldsymbol{x}_{i,1}, \boldsymbol{x}_{i,2}, \cdots, \boldsymbol{x}_{i,n_i}\}, \boldsymbol{x}_{i,j} \in \mathcal{X}, \mathcal{X} = \mathbb{R}^d, \mathcal{S}_i \subset \mathcal{Y}, \mathcal{Y} = \{l_1, l_2, \cdots, l_q\}$
$\lambda_a$ : the weight for the attention loss
$T$: the number of iterations
$\boldsymbol{X}_*$: the unseen multi-instance bag with $n_*$ instances
**Outputs**:
$\boldsymbol{Y}_*$ : the predicted label for $\boldsymbol{X}_*$
**Process**:
  1: Initialize uniform weights $\boldsymbol{w}^{(0)}$ as stated by Equation (6)
  2: **for** $t = 1$ to $T$ **do**
  3:     Shuffle training set $\mathcal{D}$ into $B$ mini-batches
  4:     **for** $b = 1$ to $B$ **do**
  5:         Extract the instance-level features of $\boldsymbol{X}$ according to Equation (1)
  6:         Calculate the attention scores as stated by Equation (2)
  7:         Map the instance-level features into a single vector representation according to Equation (3)
  8:         Update weights $\boldsymbol{w}^{(t)}$ according to Equation (7)
  9:         Calculate $\mathcal{L}$ according to Equation (8)
 10:         Set gradient $- \bigtriangledown_\Phi \mathcal{L}$
 11:         Update $\Phi$ by the optimizer
 12:     **end for**
 13: **end for**
 14: Extract the instance-level features of $\boldsymbol{X}_*$ according to Equation (9)
 15: Calculate the attention scores and map the instance-level features into a single vector representation according to Equations (10) and (11)
 16: Return $Y_*$ according to Equation (12)

---

## A.2 Additional Experiment Results

Table 5: Classification accuracy (mean±std) of each comparing algorithm on the benchmark datasets in terms of the different number of false positive labels [$r \in \{1, 2, 3\}$]. ●/○ indicates whether the performance of DEMIPL is statistically superior/inferior to the compared algorithm on each dataset (pairwise t-test at a significance level of $0.05$).

| Algorithm | $r$ | MNIST-MIPL | FMNIST-MIPL | Birdsong-MIPL | SIVAL-MIPL |
|---|---|---|---|---|---|
| DEMIPL | 1 | 0.976±0.008 | 0.881±0.021 | 0.744±0.016 | 0.635±0.041 |
| | 2 | 0.943±0.027 | 0.823±0.028 | 0.701±0.024 | 0.554±0.051 |
| | 3 | 0.709±0.088 | 0.657±0.025 | 0.696±0.024 | 0.503±0.018 |
| Mean | | | | | |
| PRODEN | 1 | 0.555±0.033● | 0.652±0.033● | 0.303±0.016● | 0.303±0.020● |
| | 2 | 0.372±0.038● | 0.463±0.067● | 0.287±0.017● | 0.274±0.022● |
| | 3 | 0.285±0.032● | 0.288±0.039● | 0.278±0.006● | 0.242±0.009● |
| RC | 1 | 0.660±0.031● | 0.697±0.166● | 0.329±0.014● | 0.344±0.014● |
| | 2 | 0.577±0.039● | 0.684±0.029● | 0.301±0.014● | 0.299±0.015● |
| | 3 | 0.362±0.029● | 0.414±0.050● | 0.288±0.019● | 0.256±0.013● |
| LWS | 1 | 0.605±0.030● | 0.702±0.033● | 0.344±0.018● | 0.346±0.014● |
| | 2 | 0.431±0.024● | 0.547±0.040● | 0.310±0.014● | 0.312±0.015● |
| | 3 | 0.335±0.029● | 0.411±0.033● | 0.289±0.021● | 0.286±0.018● |
| MaxMin | | | | | |
| PRODEN | 1 | 0.465±0.023● | 0.358±0.019● | 0.339±0.010● | 0.322±0.018● |
| | 2 | 0.338±0.031● | 0.315±0.023● | 0.329±0.016● | 0.295±0.021● |
| | 3 | 0.260±0.037● | 0.265±0.031● | 0.305±0.015● | 0.244±0.018● |
| RC | 1 | 0.518±0.022● | 0.421±0.016● | 0.379±0.014● | 0.304±0.015● |
| | 2 | 0.462±0.028● | 0.363±0.018● | 0.359±0.015● | 0.268±0.023● |
| | 3 | 0.366±0.039● | 0.294±0.053● | 0.332±0.024● | 0.244±0.014● |
| LWS | 1 | 0.457±0.028● | 0.346±0.033● | 0.349±0.013● | 0.345±0.013● |
| | 2 | 0.351±0.043● | 0.323±0.031● | 0.336±0.013● | 0.314±0.019● |
| | 3 | 0.274±0.037● | 0.267±0.034● | 0.307±0.016● | 0.268±0.019● |

Table 6: Classification accuracy (mean±std) of each comparing algorithm on the real-world dataset.

| Algorithm | CRC-MIPL-Row | CRC-MIPL-SBN | CRC-MIPL-KMeansSeg | CRC-MIPL-SIFT |
|---|---|---|---|---|
| DEMIPL | 0.408±0.010 | 0.486±0.014 | 0.521±0.012 | 0.532±0.013 |
| Mean | | | | |
| PRODEN | 0.405±0.012 | 0.515±0.010○ | 0.512±0.014● | 0.352±0.015● |
| RC | 0.290±0.010● | 0.394±0.010● | 0.304±0.017● | 0.248±0.008● |
| LWS | 0.360±0.008● | 0.440±0.009● | 0.422±0.035● | 0.338±0.009● |
| MaxMin | | | | |
| PRODEN | 0.453±0.009○ | 0.529±0.010○ | 0.563±0.011○ | 0.294±0.008● |
| RC | 0.347±0.013● | 0.432±0.008● | 0.366±0.010● | 0.204±0.008● |
| LWS | 0.381±0.011● | 0.442±0.009● | 0.335±0.049● | 0.287±0.009● |

In Section 3.2 and Section 3.3, we report the results of PRODEN, RC, and LWS using linear models. In this section, we supplement the results of the compared PLL algorithms with multi-layer perceptrons (MLPs) as described in the respective literature. Table 5 and Table 6 present the experimental results of DEMIPL compared to PLL algorithms on the benchmark and real-world datasets, respectively. It is noteworthy that DEMIPL employs a two-layer CNN network for feature extraction on the MNIST-MIPL and FMNIST-MIPL datasets, while on the remaining datasets, DEMIPL only utilizes linear models.

On the benchmark datasets, DEMIPL consistently outperforms the compared algorithms in almost all cases. Moreover, the compared algorithms using MLPs do not consistently yield superior results compared to those using linear models, especially when the benchmark datasets exhibit relatively simple features. This suggests that linear models are sufficient to achieve satisfactory results given the benchmark datasets, while MLPs might introduce unnecessary complexity.

On the real-world dataset, DEMIPL outperforms the compared algorithms in 19 out of 24 cases. When combined with complex image bag generators such as CRC-MIPL-KMeansSeg and CRC-MIPL-SIFT, DEMIPL outperforms the compared algorithms in 11 out of 12 cases. In the majority of cases, the compared algorithms using MLPs demonstrate better performance than those using linear models.

However, on the CRC-MIPL-SIFT dataset, the improvement provided by MLPs is not particularly evident and sometimes even leads to a decline in performance. Therefore, when dealing with complex multi-instance features, the bag features obtained through the Mean or MaxMin strategies do not accurately reflect the characteristics of multi-instance bags. This highlights the need for specialized MIPL algorithms to accurately capture the features of multi-instance bags.

## A.3 Theoretical Analysis

**Theorem 1.** *In a multi-instance bag $\boldsymbol{X}_i$, when the normalized attention score of an instance $\boldsymbol{x}_{i,j'}$ approaches $1$, e.g., $\frac{a_{i,j'}}{\sum_{j=1}^{n_i} a_{i,j}} \to 1$, the probability of the multi-instance bag $\boldsymbol{X}_i$ being classified as the c-th class is approximately equal to that of the instance $\boldsymbol{x}_{i,j'}$ belonging to the c-th class.*

*Proof.* Equation (2) demonstrates that the attention score for each instance ranges between $0$ and $1$. After normalizing by $\frac{1}{\sum_{j=1}^{n_i} a_{i,j}}$, the sum of attention scores for all instances within a multi-instance bag becomes equal to $1$. When the normalized attention score of an instance $\boldsymbol{x}_{i,j'}$ is approach 1, the normalized attention scores of the remaining instances $\{\boldsymbol{x}_{i,1}, \boldsymbol{x}_{i,2}, \cdots, \boldsymbol{x}_{i,n_i}\} \setminus \{\boldsymbol{x}_{i,j'}\}$ approach 0. Based on Equation (3), the aggregated bag-level vector representation $\boldsymbol{z}_i = \frac{1}{\sum_{j=1}^{n_i} a_{i,j}} \sum_{j=1}^{n_i} a_{i,j} \boldsymbol{h}_{i,j} \approx \boldsymbol{h}_{i,j'} = h(\boldsymbol{x}_{i,j'})$. Therefore, it is confirmed that instances with higher attention scores contribute significantly to the bag-level predictions, underlining the significance of attention mechanisms in multi-instance partial-label learning. $\square$

Theorem 1 suggests that high attention scores of individual instances can play a crucial role in determining the bag-level class prediction, emphasizing the significance of accurately capturing and interpreting attention scores in multi-instance partial-label learning scenarios.

## A.4 Image Bag Generator

We utilize four image bag generators to extract multi-instance features from the CRC-MIPL dataset. The detailed descriptions of these image bag generators are provided below:

- **Row Generator**: It treats each row of the image as an individual instance. To extract the feature for each instance, the Row generator computes the average RGB color value of the row and the color differences in the rows above and below it.
- **SBN Generator**: It considers each $2 \times 2$ blob within the image and includes the RGB color values of the blob itself and its four neighboring blobs as features for each instance. It generates instances by iteratively moving one pixel at a time. However, it should be noted that the SBN generator ignores the feature information at the four corners of the image.
- **KMeansSeg Generator**: It divides the image into $K$ segments or partition blocks. For each segment, it generates a 6-dimensional feature. The first three dimensions represent color values in the YCbCr color space, while the last three dimensions represent values obtained by applying the wavelet transform to the luminance (Y) component of the image.
- **SIFT Generator**: It applies the scale-invariant feature transform (SIFT) algorithm to extract features, which partitions each instance into multiple $4 \times 4$ subregions and assigns the gradients of the pixels within these subregions to 8 bins. Consequently, the SIFT generator produces a 128-dimensional feature vector for each instance.

The implementations of the four image bag generators are available at `http://www.lamda.nju.edu.cn/code_MIL-BG.ashx`.

## A.5 Data and Code Availability

The implementations of the compared algorithms are publicly available. MIPLGP and PL-AGGD are implemented at `http://palm.seu.edu.cn/zhangml/`. PRODEN is implemented at `https://github.com/Lvcrezia77/PRODEN`. RC is implemented at `https://lfeng-ntu.github.io/codedata.html`. LWS is implemented at `https://github.com/hongwei-wen/LW-loss-for-partial-label`. Additionally, the code of DEMIPL, the benchmark datasets, and the real-world dataset are publicly available at `http://palm.seu.edu.cn/zhangml/`.

