# OpenReview forum: "Disambiguated Attention Embedding for Multi-Instance Partial-Label Learning"
_NeurIPS.cc/2023/Conference — NeurIPS 2023 poster_

### Official Review · Reviewer_hzaM · 2023-06-28

**Soundness:** 3 good
**Presentation:** 4 excellent
**Contribution:** 4 excellent
**Rating:** 8
**Confidence:** 4

**Summary:**

This paper proposes a new algorithm for a weakly supervised learning paradigm called multi-instance partial label learning, where each sample is represented by a multi-instance bag and its label is provided as a set of candidate labels. This is a particularly challenging learning paradigm since learners do not have access to which label is the ground truth label from the candidate label set, and do not know which instance corresponds to the ground truth label.  The authors proposed a deep learning algorithm to solve the problem integrating the attention mechanism with a disambiguation strategy. The attention mechanism map the multi-instance representation into a single-vector and the disambiguation strategy identifies the true label from the candidate label set. The synergy of these two components is validated with a wide range of benchmark datasets. Furthermore, this manuscript also proposed a new crowd-labelled medical dataset that can be used as future benchmarks for this learning task.

**Strengths:**

* The problem setting that motivates this study is novel and potentially impactful. The multi-instance partial label learning paradigm simultaneously addresses the challenges two of the most widely studied weakly supervised learning setting, multi-instance learning and partial label learning.
* The paper introduces a new real-world dataset for histopathology image classification from weak supervision in the form of multi-instance bags supervised with candidate label sets. This indicates that the proposed learning paradigm and algorithm can potentially alleviate the labelling cost of medical image dataset.
* The proposed disambiguation attention mechanism applies to both binary and multi-class classification scenarios, whereas existing attention mechanisms designed for multi-instance bags can only handle binary classification.
* The proposed algorithm demonstrates good performance improvement compared to a wide range of existing arts.


**Weaknesses:**

* The synergy of disambiguation attention mechanism and momentum-based disambiguation strategy can be further explored/evaluated. See questions for more on this.

**Questions:**

* DEMIPL performs classification by aggregates the instances into a single-vector bag representation. Would it be better if instance representation and bag representation are used simultaneously for classification and disambiguation?
* Since the momentum-based disambiguation strategy degenerates to progressive disambiguation strategy proposed in [39], how would the performance be if momentum-based disambiguation is used directly in the algorithm proposed in [39]? This may provide empirical evidence on the synergy of momentum-based disambiguation and disambiguation attention.
* Is there feasibility to learn which instance corresponds to the true label in the candidate label set, i.e., instance-level classification from MIPL datasets?
* Is there any plan for releasing the CRC-MIPL dataset?


**Limitations:**

This paper does not explicitly discuss its limitations.

---

> ### Author Rebuttal · Authors · 2023-08-09
>
> We greatly appreciate your constructive feedback and comments, which have proven instrumental in enhancing the quality of our paper. Your recognition of the positive aspects of our work is sincerely valued. In the subsequent sections, we present your comments and our responses in a point-by-point manner.
>
> > The synergy of disambiguation attention mechanism and momentum-based disambiguation strategy can be further explored/evaluated.
>
> Below, we elaborate further on key points:
>
> The MIPL dataset inherently features dual inexact supervision. The presence of inexact supervision within multi-instance bags directly impacts the disambiguation process, while the inexact supervision associated with candidate label sets influences the aggregation of features at the bag level. First, the disambiguation attention mechanism plays a pivotal role in aggregating multi-instance bags into a coherent bag-level feature representation. Precise predictive probabilities contribute significantly to refining the accuracy of the aggregated bag-level features. Second, the classifier employs this feature representation as input, generating predictive probabilities for the given multi-instance bag. Accurate bag-level features markedly enhance the precision of the predictive probabilities. Last, the comprehensive loss function entails a weighted summation of losses from both the disambiguation attention mechanism and the momentum-based disambiguation strategy. Any failure of either of the losses to converge could potentially hamper the performance of the other component. The model attains convergence solely when both these losses converge. Consequently, we posit that the disambiguation attention mechanism and the momentum-based disambiguation strategy exhibit a synergistic interplay.
>
> > DEMIPL performs classification by aggregates the instances into a single-vector bag representation. Would it be better if instance representation and bag representation are used simultaneously for classification and disambiguation?
>
> This comment presents an intriguing perspective. We hypothesize that a potential avenue lies in effectively leveraging both instance and bag representations concurrently, leading to a valuable integrated representation for classification and disambiguation purposes. Nonetheless, the challenge lies in identifying such an effective approach; otherwise, the risk of overfitting remains a concern.
>
>
> > Since the momentum-based disambiguation strategy degenerates to progressive disambiguation strategy proposed in [39], how would the performance be if momentum-based disambiguation is used directly in the algorithm proposed in [39]? This may provide empirical evidence on the synergy of momentum-based disambiguation and disambiguation attention.
>
> We replace the progressive disambiguation strategy in PRODEN with the momentum-based disambiguation strategy, which we refer to as PRODEN-MD. Both PRODEN and PRODEN-MD utilize the parameter recommendations outlined in [39]. Aside from the disambiguation strategy, no other distinctions exist between the two approaches. For the MNIST, FMNIST, and KMNIST datasets, we set the flipping probabilities $q$ to $\\{0.1, 0.3, 0.5, 0.7, 0.9\\}$ and perform $5$ runs for each configuration. The resulting mean accuracy and standard deviation are presented in the table below.
>
> | Dataset | Algorithm |      $q=0.1$      |      $q=0.3$      |      $q=0.5$      |      $q=0.7$      |      $q=0.9$      |
> | ----- | ------- | :---------------: | :---------------: | :---------------: | :---------------: | :---------------: |
> |  MNIST  |  PRODEN   | $0.926 \pm 0.000$ | $0.925 \pm 0.001$ | $0.923 \pm 0.001$ | $0.916 \pm 0.001$ | $0.900 \pm 0.002$ |
> |         | PRODEN-MD | $0.927 \pm 0.001$ | $0.925 \pm 0.001$ | $0.923 \pm 0.001$ | $0.918 \pm 0.001$ | $0.902 \pm 0.002$ |
> | FMNIST  |  PRODEN   | $0.843 \pm 0.001$ | $0.843 \pm 0.001$ | $0.840 \pm 0.002$ | $0.834 \pm 0.002$ | $0.818 \pm 0.003$ |
> |         | PRODEN-MD | $0.844 \pm 0.000$ | $0.842 \pm 0.001$ | $0.840 \pm 0.002$ | $0.835 \pm 0.001$ | $0.820 \pm 0.002$ |
> | KMNIST  |  PRODEN   | $0.695 \pm 0.002$ | $0.691 \pm 0.002$ | $0.681 \pm 0.003$ | $0.668 \pm 0.005$ | $0.632 \pm 0.003$ |
> |         | PRODEN-MD | $0.697 \pm 0.001$ | $0.691 \pm 0.002$ | $0.680 \pm 0.003$ | $0.666 \pm 0.004$ | $0.639 \pm 0.003$ |
>
> The results indicate that, in most instances, PRODEN-MD achieves a slightly higher average accuracy compared to PRODEN, albeit with a negligible difference. We attribute this phenomenon to the presence of dual inexact supervision in MIPL data, in contrast to PLL data which only contains inexact supervision in the output space. Given the lower learning complexity of PLL compared to MIPL, the progressive disambiguation strategy suffices to achieve satisfactory performance in disambiguation. Additionally, this underscores the significant performance improvement achieved through the synergistic utilization of the momentum-based disambiguation strategy and the attention mechanism. Your insights are greatly appreciated.
>
>
>
> > Is there feasibility to learn which instance corresponds to the true label in the candidate label set, i.e., instance-level classification from MIPL datasets?
>
> We posit the feasibility of acquiring an instance-level classifier from MIPL datasets. Our forthcoming endeavors will encompass the design of a dedicated instance-level classifier aimed at predicting instance-level labels, facilitating the identification of instances aligned with ground-truth labels.
>
>
>
> > Is there any plan for releasing the CRC-MIPL dataset?
>
> Upon acceptance of the paper, we will make the CRC-MIPL dataset and the DEMIPL code publicly available as open-source.
>
> References:
>
> [39] Jiaqi Lv, Miao Xu, Lei Feng, Gang Niu, Xin Geng, and Masashi Sugiyama. Progressive identification of true labels for partial-label learning. In Proceedings of the 37th International Conference on Machine Learning, Virtual Event, pages 6500–6510, 2020.

---

> > ### Comment · Reviewer_hzaM · 2023-08-20
> >
> > Thanks for the author's response. I would also maintain my initial score of accept.

---

> > > ### Author Response · Authors · 2023-08-20
> > > **Thank you**
> > >
> > > We appreciate your score and prompt response.

---

### Official Review · Reviewer_7xX7 · 2023-06-28

**Soundness:** 2 fair
**Presentation:** 3 good
**Contribution:** 2 fair
**Rating:** 5
**Confidence:** 3

**Summary:**

The paper proposes the first deep learning-based method for multi-instance partial-label learning (MIPL) problem. The proposed method employs a disambiguation attention mechanism to approximate the contribution of each instance to the bag-level feature and then aggregates a multi-instance bag into a single vector representation, with an attention loss introduced to enhance the discriminativeness between positive and negative instances. The method then uses the extracted feature to identify the ground-truth label from the candidate label set using a momentum-based disambiguation strategy and simultaneously learns the classifier. Furthermore, the paper introduces a real-world MIPL dataset for colorectal cancer classification. Empirical results confirm the effectiveness of the proposed method and the contribution of the key designs included.

**Strengths:**

1.	The paper is reasonably clear and comprehensive.
2.	The paper’s claims are supported by experimental and ablative results.


**Weaknesses:**

1.	A notation in Section 2.1 at line 91 is incorrect. Since none of the instances in the bag belong to any of the false positive labels as the author claims in line 42, the first \mathcal{Y} at line 91 should be replaced to \boldsymbol{S}_{i}.
2.	I don’t think the weights calculated by equation (2) is “attention” scores, since the computation process only involves a single instance without interaction between different instances, and there does not exist query, key and value, or they are all the same instance. These scores are only weights that approximate the contribution of different instances to the bag.
3.	I have some doubts about the proposed method. See the question part below.


**Questions:**

1.	I have questions about why does the method work. The momentum-based disambiguation strategy simply minimizes the cross entropy loss between the predicted label and all candidate labels, with importance weights added which are obtained by momentum updates from probability prediction. I wonder why this method ensure that the positive label in the candidate set to be correctly selected (disambiguated) ? If this does not give a correct direction, the representation of the instance bag will also be misled, and how can the model reach satisfactory state when converge as the author claims in line 158 under this situation?
2.	The method proposes to minimize the “entropy” of the attention weights calculated to make the attention scores between positive and false positive labels discriminative enough. However, if multiple instances in the instance bag share this positive label, I worry that simply minimizing the entropy would result in a single instance with high attention score against all other low-score instances, neglecting the other positive instances, so the consistency between attention scores and instance-level labels is broken.

---

> ### Author Rebuttal · Authors · 2023-08-09
>
> Thank you for dedicating your time to reading and reviewing our paper. Your comments have provided valuable insights that greatly contribute to the enhancement of our work. In the following sections, we present your comments and provide detailed responses to each of them.
>
> > A notation in Section 2.1 at line 91 is incorrect. Since none of the instances in the bag belong to any of the false positive labels as the author claims in line 42, the first \mathcal{Y} at line 91 should be replaced to \boldsymbol{S}_{i}.
>
> We have diligently addressed the issue highlighted in your comment and conducted a comprehensive review of the entire paper.
>
>
>
> > I don’t think the weights calculated by equation (2) is “attention” scores, since the computation process only involves a single instance without interaction between different instances, and there does not exist query, key and value, or they are all the same instance. These scores are only weights that approximate the contribution of different instances to the bag.
>
> In the field of multi-instance learning (MIL), pioneering work by [7] introduced attention mechanisms to consolidate multi-instance bags into single feature representations, where the query, key, and value are all identical. We extend this concept by introducing the disambiguation attention mechanism and conventionally refer to the result of Equation (2) as attention scores. These scores offer insights into the contributions of each instance to the bag's overall representation. We have incorporated these modifications in the revised version.
>
> >  I have questions about why does the method work. The momentum-based disambiguation strategy simply minimizes the cross entropy loss between the predicted label and all candidate labels, with importance weights added which are obtained by momentum updates from probability prediction. I wonder why this method ensure that the positive label in the candidate set to be correctly selected (disambiguated) ? If this does not give a correct direction, the representation of the instance bag will also be misled, and how can the model reach satisfactory state when converge as the author claims in line 158 under this situation?
>
> The empirical risk of MIPL can be expressed as $\mathcal{R}(f)=\mathbb{E}[\mathcal{L}\_{MIPL}(\boldsymbol{X}\_i, \boldsymbol{S}\_i)]$. To estimate the empirical risk, an intuitive approach is to define $\mathcal{L}\_{MIPL}(\boldsymbol{X}\_i, \boldsymbol{S}\_i)= \text{min}\_{Y\_i \in \boldsymbol{S}\_i}\mathcal{L}(\boldsymbol{X}\_i, Y\_i)$. However, straightforwardly optimizing the above empirical risk has two drawbacks: firstly, optimizing the $\min$ operator is intricate, and secondly, if an incorrect label is chosen initially, subsequent iterations will also be erroneous. To address these issues, we propose to assign higher weights to labels with lower loss values.
>
> Due to the existence of dual inexact supervision in multi-instance partial-label learning (MIPL), the learning challenge surpasses that of partial-label learning (PLL). If only the current round's model output is used as weights, the disambiguation performance would deteriorate when confronted with an increased number of false positive labels. Therefore, in order to enhance the disambiguation effectiveness of our algorithm under conditions of a substantial number of false positive labels, we propose using a weighted sum of previous and current round prediction probabilities as weights. The strategy facilitates a measure of correction within the iterative process, even in cases where exact labels are not initially discerned. Experimental results in Figure 4 demonstrate the superior effectiveness of the momentum-based disambiguation strategy compared to directly using the current round's output as weights or disregarding weight updates.
>
> Since the attention mechanism and the classifier exert reciprocal influence on each other throughout the training process, model convergence occurs exclusively when both of their respective loss values are minimized. Thus, we posit that a state of satisfaction is attained when the individual loss values of both components independently converge.
>
>
> > The method proposes to minimize the “entropy” of the attention weights calculated to make the attention scores between positive and false positive labels discriminative enough. However, if multiple instances in the instance bag share this positive label, I worry that simply minimizing the entropy would result in a single instance with high attention score against all other low-score instances, neglecting the other positive instances, so the consistency between attention scores and instance-level labels is broken.
>
> The remarks are insightful, and we have indeed incorporated this consideration during the design of our algorithm. While the attention scores calculated using Equation (2) are bounded within the interval $[0, 1]$, our implementation does not enforce the sum of attention scores across all instances within a bag to equate to $1$. Therefore, we can ensure that, while minimizing the entropy of the attention scores, large attention weights can be assigned to multiple positive instances within a multi-instance bag.
>
> References:
>
> [7] Maximilian Ilse, Jakub M. Tomczak, and Max Welling. Attention-based deep multiple instance learning. In Proceedings of the 35th International Conference on Machine Learning, Stockholmsmässan, Stockholm, Sweden, pages 2132–2141, 2018.

---

### Official Review · Reviewer_suFM · 2023-07-03

**Soundness:** 3 good
**Presentation:** 3 good
**Contribution:** 3 good
**Rating:** 7
**Confidence:** 3

**Summary:**

The article proposes a novel algorithm called DEMIPL for addressing the challenges in multi-instance partial-label learning (MIPL) tasks. The algorithm uses a disambiguation attention mechanism to embed multi-instance bags into a single vector representation and a momentum-based disambiguation strategy to identify the ground-truth label from candidate label sets. Experimental results on benchmark and real-world datasets demonstrate the superiority of DEMIPL over existing MIPL approaches.

**Strengths:**

1. The paper is easy to read and the core idea is clearly presented.
2. DEMIPL proposes a new algorithm that addresses the limitations of existing MIPL approaches and introduces a disambiguation attention mechanism and momentum based disambiguation strategy.
3. The article presents experimental results on benchmark and real-world datasets, demonstrating the superiority of DEMIPL over other approaches.

**Weaknesses:**

1. Recently,  using distribution of labels to generate partial labels have been considered as more difficult data scenario,  which  increases the difficulty for the model to identify similar samples. The authors are  encoraged to  incorporate such type of data for experiments.
2. The reason why Eq. 7 uses model prediction confidence and weights in previous training rounds needs to be further clearly clarified.

**Questions:**

1. In addition to the MNIST-MIPL and FMNIST-MIPL data sets, what is the input of other data sets to attention, are the features alreadly extracted?
2. Comparing the attention-based network with the comparison method, is there any unfairness?

**Limitations:**

See above questions.

---

> ### Author Rebuttal · Authors · 2023-08-09
>
> Thank you for providing constructive feedback and insightful comments to enhance our paper. We greatly value your recognition of the positive aspects of our work. We will now address the specific concerns you raised.
>
> > Recently, using distribution of labels to generate partial labels have been considered as more difficult data scenario, which increases the difficulty for the model to identify similar samples. The authors are encoraged to incorporate such type of data for experiments.
>
> The approach you mention for generating partial labels bears a resemblance to instance-dependent partial-label learning [23], which generates partial labels for benchmark datasets based on predicted label distributions. Our proposed CRC-MIPL dataset comprises $7$ categories, each containing $1000$ images. Noteworthy similarities exist both within individual images and in their candidate label sets. Consequently, experimental results on the CRC-MIPL dataset can effectively showcase DEMIPL's proficiency in handling more intricate scenarios. We appreciate your astute suggestion and have plans to create dedicated datasets in the future.
>
>
> > The reason why Eq. 7 uses model prediction confidence and weights in previous training rounds needs to be further clearly clarified.
>
> First, we expound upon the necessity for weight updates. Subsequently, we elucidate the motivation behind employing Equation (7) for weight updates.
>
> In the realm of partial-label learning (PLL), the empirical superiority of PRODEN over PRODEN-naive validates the practice of updating weights, which typically yields improved outcomes in contrast to scenarios where weights remain unchanged [39]. This observation extends to multi-instance partial-label learning (MIPL) as well. As illustrated in Figure 4, the classification accuracy of DEMIPL and DEMIPL-PR surpasses that of DEMIPL-AV.
>
> Furthermore, MIPL data involves dual inexact supervision. The presence of inexact supervision in multi-instance bags affects the disambiguation, and inexact supervision within the candidate label set influences the aggregation of bag-level features. As the count of false positive labels increases, the accuracy of prediction confidence diminishes. Therefore, directly updating weights using low-confidence predictions can pose challenges. Instead, we choose to gradually calibrate the model's output by combining the prediction confidence of the current round with that of the previous round. As training rounds progress, the model's predictions gradually become more accurate, leading to a decrease in the weight of the previous round's prediction confidence. This weight update strategy effectively enhances classification accuracy, particularly when dealing with a substantial number of false positive labels.
>
> For these reasons, we adopt the weighted sum of prediction confidence from the current round and the previous round as the new weight.
>
>
>
> > In addition to the MNIST-MIPL and FMNIST-MIPL data sets, what is the input of other data sets to attention, are the features alreadly extracted?
>
> Yes. The benchmark datasets are curated and made openly available in [30]. For the MNIST-MIPL and FMNIST-MIPL datasets, we utilize a convolutional neural network with two convolutional layers to extract primitive features of size $28 \times 28 \times 1$. The Birdsong and SIVAL datasets have their features extracted and released by [42] and [43], respectively. These datasets are transformed into MIPL datasets in [30], namely Birdsong-MIPL and SIVAL-MIPL. To leverage the pre-extracted features, we employ a fully connected network.
>
>
>
> > Comparing the attention-based network with the comparison method, is there any unfairness?
>
> We believe that there is no inherent unfairness in our approach. To the best of our knowledge, DEMIPL represents the second algorithm in this field. Currently, the only existing MIPL algorithm is MIPLGP [30]. Given that PLL algorithms are incapable of directly handling MIPL data, we are compelled to transform the MIPL data into PLL data using the Mean and Maxmin degradation strategies. Initially, we attempted to employ a multi-instance bag's candidate label set for each individual instance within the bag. Unfortunately, due to the lack of numerous ground-truth labels for negative instances within the candidate label set, the outcomes of the PLL algorithm proved unsatisfactory. In the future, as more MIPL algorithms become available, our primary focus will be on comparative evaluations against these algorithms.
>
> References:
>
> [23] Ning Xu, Congyu Qiao, Xin Geng, and Min-Ling Zhang. Instance-dependent partial label learning. Advances in Neural Information Processing Systems 34, Virtual Event, pages 27119–393 27130, 2021
>
> [30] Wei Tang, Weijia Zhang, and Min-Ling Zhang. Multi-instance partial-label learning: Towards exploiting dual inexact supervision. Science China Information Sciences, pages 1–16, 2023.
>
> [39] Jiaqi Lv, Miao Xu, Lei Feng, Gang Niu, Xin Geng, and Masashi Sugiyama. Progressive identification of true labels for partial-label learning. In Proceedings of the 37th International Conference on Machine Learning, Virtual Event, pages 6500–6510, 2020.
>
> [42] Forrest Briggs, Xiaoli Z. Fern, and Raviv Raich. Rank-loss support instance machines for MIML instance annotation. In Proceedings of the 18th ACM SIGKDD International Conference on Knowledge Discovery and Data Mining, Beijing, China, pages 534–542, 2012.
>
> [43] Burr Settles, Mark Craven, and Soumya Ray. Multiple-instance active learning. In Advances in Neural Information Processing Systems 20, Vancouver, British Columbia, Canada, pages 1289–1296, 2007.

---

> > ### Comment · Reviewer_suFM · 2023-08-19
> >
> > Thanks for the authors' thorough rebuttal that fully addressed my concerns. I have no major concerns.

---

> > > ### Author Response · Authors · 2023-08-19
> > > **Thank you**
> > >
> > > Thank you for your response and the improved rating. We are delighted to have addressed your major concerns.

---

### Official Review · Reviewer_EhJn · 2023-07-06

**Soundness:** 3 good
**Presentation:** 3 good
**Contribution:** 2 fair
**Rating:** 7
**Confidence:** 4

**Summary:**

The paper introduces a novel algorithm named Disambiguated attention Embedding for Multi-Instance Partial-Label learning (DEMIPL) to address the problem of multi-instance partial-label learning (MIPL). The proposed algorithm embeds multi-instance bags into a single vector representation using a disambiguation attention mechanism and a momentum-based disambiguation strategy. The paper also presents a real-world MIPL dataset for colorectal cancer classification. Experimental results demonstrate the superiority of DEMIPL over existing MIPL and partial-label learning approaches.

**Strengths:**

1. DEMIPL introduces a unique scheme of embedding multi-instance bags into a single vector representation, addressing the limitations of existing MIPL approaches.
2. Disambiguation Attention: The use of a disambiguation attention mechanism allows for better aggregation of bag-level information, considering global context, and reducing sensitivity to negative instance predictions.
3. Momentum-Based Disambiguation: DEMIPL employs a momentum-based strategy to identify the ground-truth label, improving the accuracy of label prediction.
4. The paper introduces a real-world MIPL dataset for colorectal cancer classification, enabling evaluation in a practical setting. Experimental results demonstrate that DEMIPL outperforms other established MIPL and partial-label learning approaches, showcasing its effectiveness in solving the MIPL problem.

**Weaknesses:**

1. When discussing the experimental results, provide more information about the benchmark datasets used for evaluation. Explain the characteristics and sizes of the datasets, as well as any preprocessing steps applied. Additionally, describe the performance metrics used to assess the classification accuracy and provide a clear comparison with existing MIPL and partial-label learning approaches.
2. While the proposed DEMIPL algorithm demonstrates superior performance compared to existing approaches, it is important to acknowledge any limitations of the proposed method. Discuss potential scenarios or datasets where DEMIPL may not perform optimally and propose directions for future research to address these limitations.
3. Proofread the paper for grammatical errors and improve the overall clarity of the writing.
4. Consider providing more details about the real-world MIPL dataset for colorectal cancer classification. Describe the acquisition process of the dataset, the number of images per class, and any challenges or biases associated with the dataset.
5. Consider providing a high-level visual representation or diagram of the DEMIPL algorithm to aid in understanding the proposed approach.
6. Ensure consistency in notation and symbols throughout the paper.

**Questions:**

See the above.

**Limitations:**

See the above.

---

> ### Author Rebuttal · Authors · 2023-08-09
>
> Thank you for your valuable feedback and insightful suggestions to enhance the quality of our manuscript. We have diligently integrated your input into the revised version. In the following sections, we present your comments, followed by our detailed responses.
>
> > When discussing the experimental results, provide more information about the benchmark datasets used for evaluation. Explain the characteristics and sizes of the datasets, as well as any preprocessing steps applied. Additionally, describe the performance metrics used to assess the classification accuracy and provide a clear comparison with existing MIPL and partial-label learning approaches.
>
> Due to space constraints, we provided a concise overview of the benchmark datasets in Section 3.1. For a more comprehensive exposition, encompassing processing steps, we have elaborated on these aspects in the Appendix.
>
> The performance metric employed in multi-instance partial-label learning (MIPL) is the conventional classification accuracy for multi-classification, defined as $\text{accuracy}=\frac{1}{m}\sum_{i=1}^{m}\mathbb{I}(f(\boldsymbol{z}_i), \boldsymbol{Y}_i)$, where $m$ denotes the number of multi-instance bags in the test dataset, and $\boldsymbol{z}_i$ and $\boldsymbol{Y}_i$ signify the aggregated feature representation and the ground-truth label of the $i$-th multi-instance bag.
>
> Regarding the MIPLGP algorithm, we utilize the same dataset as DEMIPL and adopt the parameter settings recommended in [30]. As partial-label learning (PLL) algorithms are not inherently suited for MIPL data, we apply Mean and Maxmin strategies to transform MIPL datasets into PLL-compatible ones (lines 205-209). Moreover, we have augmented the revised version with further detailed comparisons and in-depth analyses of divergent outcomes.
>
>
> > It is important to acknowledge any limitations of the proposed method. Discuss potential scenarios or datasets where DEMIPL may not perform optimally and propose directions for future research to address these limitations.
>
>
> In the Appendix, we have attached limitations as follows:
>
> DEMIPL exhibits certain limitations, and there are several unexplored directions in the realm of MIPL. For example, DEMIPL assumes independence among instances within each bag. A promising avenue for future research involves considering dependencies between instances during aggregation. Moreover, akin to multiple-instance learning (MIL) algorithms grounded in the embedded-space paradigm, accurately predicting instance-level labels poses a challenging endeavor. One possible approach entails the introduction of an instance-level classifier. Furthermore, DEMIPL operates under the assumption that the ground-truth label of each multi-instance bag resides within the candidate label set. In the future, relaxing this assumption could encompass scenarios where some ground-truth labels of multi-instance bags lie outside the candidate label set.
>
>
> > Proofread the paper for grammatical errors and improve the overall clarity of the writing.
>
> We have thoroughly and persistently proofread the paper to rectify any grammatical concerns and ensure that the content is presented with utmost clarity.
>
>
> > Consider providing more details about the real-world MIPL dataset for colorectal cancer classification. Describe the acquisition process of the dataset, the number of images per class, and any challenges or biases associated with the dataset.
>
> Due to limited space, we only offered a concise overview of the real-world CRC-MIPL dataset in Section 3.1 and Appendix A.5. We now provide additional comprehensive details about the CRC-MIPL dataset, which will be added to the revised manuscript.
>
> We select $7000$ images from the NCT-CRC-HE-100K dataset [44] for colorectal cancer classification, with $1000$ images per class. Instance-level features for each image are obtained using four image bag generators, while the candidate label set for each image is derived from the annotations of three crowdsourcing workers. The sample distribution is balanced across classes. The dataset poses significant challenges due to the lack of exact label information, and approximately $1.5\\%$ of ground-truth labels are not included in the candidate label set.
>
> > Consider providing a high-level visual representation or diagram of the DEMIPL algorithm to aid in understanding the proposed approach.
>
> We will incorporate your suggestion by including a high-level visual representation or diagram of the DEMIPL algorithm in the revised version. The high-level procedure of DEMIPL can be described as follows: given a histopathological image along with its corresponding candidate label set, we aim to provide a clear visual overview. The following outlines the succinct workflow. Initially, we employ an image bag generator to extract instance-level features. Subsequently, an attention mechanism, implemented through a fully connected network, calculates attention scores. These attention scores, in conjunction with the instance-level features, are used to aggregate a bag-level feature representation. Finally, a classifier is employed to predict the class probabilities of the bag-level features.
>
> > Ensure consistency in notation and symbols throughout the paper.
>
> We have conducted thorough and meticulous reviews of the entire paper, implementing essential revisions to ensure consistency in notation and symbols. For instance, we have replaced $\boldsymbol{S}$ with $\mathcal{S}$ to denote the candidate label set and modified Equation (4) to $\mathcal{L}\_a = - \sum\_{i=1}^{m}\sum_{j=1}^{n_i} a_{i,j} \log a_{i,j}$.
>
> References:
>
> [30] Wei Tang, Weijia Zhang, and Min-Ling Zhang. Multi-instance partial-label learning: Towards exploiting dual inexact supervision. Science China Information Sciences, pages 1–16, 2023.
>
> [44] Jakob Nikolas Kather, et al. Predicting survival from colorectal cancer histology slides using deep learning: A retrospective multicenter study. PLoS Medicine, 16(1):e1002730:1–22, 2019.

---

> > ### Comment · Reviewer_EhJn · 2023-08-15
> > **I am satisfied with your response.**
> >
> > Thank you for your response. I would change the rating to accept.

---

> > > ### Author Response · Authors · 2023-08-15
> > > **Thank you**
> > >
> > > Thank you for raising your score.

---

### Official Review · Reviewer_pq8x · 2023-07-07

**Soundness:** 3 good
**Presentation:** 3 good
**Contribution:** 2 fair
**Rating:** 5
**Confidence:** 3

**Summary:**

In this paper the authors study multi-instance partial label learning, where a bag of instance is associated with several candidate labels of which it is assumed one is correct. The propose a deep learning approach for such a problem. The key contribution is a "disambiguation attention mechanism" where they find the relevance of each instance to each class. Each bag is then represented by a weighted average of instance features with the attention being the weights. They modify the loss function to improve consistency between attention scores and instance labels. Further they also propose a momentum based loss for the labels that gives higher weight to higher class probabilities. The provide experimental results on several synthetic datasets and one "real-world" dataset.

**Strengths:**

+ the MIPL setting could have some applications in the real world.
+ Using attention to construct bag features seems like an interesting idea.
+ a variety of experimental results have been reported against several baselines.
+ additional experiments study the effect of the contributions in isolation.

**Weaknesses:**

- The assumption of a single correct label seems very strong and unrealistic in practice.
- The histopathology task seems really contrived. It is hard to imagine non-experts labeling such images, and if they did, how could it be assumed any labels were correct?
- Without a compelling application, it is not clear what is the significance of this work.

**Questions:**

How was label quality controlled in the histo dataset to ensure there was one correct label?

**Limitations:**

not adequately discussed

---

> ### Author Rebuttal · Authors · 2023-08-09
>
> Thanks for your valuable feedback and insightful comments, which have greatly contributed to the enhancement of our paper. We also appreciate your acknowledgment of the positive attributes of our work. We will now address the mentioned negative aspects. Below, we provide a summary of your comments along with our corresponding responses.
>
> > The assumption of a single correct label seems very strong and unrealistic in practice.
>
> The assumption that the candidate label set contains a single correct label has been extensively investigated in partial-label learning (PLL). Unlike PLL, the task of multi-instance partial-label learning (MIPL) is more intricate and is still in its initial stages of development. To the best of our knowledge, the only existing work on MIPL is [30], which shares this assumption. It is noteworthy that this assumption holds true for many scenarios. For instance, in the CRC-MIPL dataset, over $98.5\\%$ of ground-truth labels for multi-instance bags are indeed encompassed within their respective candidate label sets. The primary objectives of this paper are twofold: the proposal of an algorithm based on the embedded-space paradigm and the introduction of a real-world MIPL dataset. In future research, we intend to relax the assumption that candidate label sets invariably contain the true label and aim to devise effective algorithms.
>
>
> > It is hard to imagine non-experts labeling such images, and if they did, how could it be assumed any labels were correct? How was label quality controlled in the histo dataset to ensure there was one correct label? Without a compelling application, it is not clear what is the significance of this work.
>
> Based on our investigation, the utilization of non-expert annotations for histopathological images has been established as a viable approach, and crowdsourcing amateur labelers can achieve a good level of accuracy [a, b]. CRC-MIPL is derived from a larger dataset NCT-CRC-HE-100K used for colorectal cancer classification [44]. We select $1000$ images from each of the seven classes. Before annotating the CRC-MIPL dataset, we invite an expert to provide training to three crowdsourcing workers. During the labeling process, we shuffle the order of the images, and the ground-truth labels are hidden from the crowdsourcing workers. Each worker annotates a candidate label set for each image, and the final candidate label set is obtained by aggregating the candidates from all three workers. The detailed methodology is outlined as follows: First, each of the workers assigns candidate labels with non-zero probabilities to form a label set per image. A higher probability indicates a higher likelihood of being the ground-truth label, while a probability of zero implies the label is a non-candidate label. Second, after obtaining three label sets for each image, we derive the final candidate label set through the following procedure. A label present in two or three label sets is included in the final candidate label set. If the final candidate label set comprises only one or no label, we select the labels corresponding to the highest probability in each label set.
>
> Utilizing this procedure, the candidate label sets for more than $98.5\\%$ of the samples contain the ground truth labels in the CRC-MIPL dataset. Compared to involving experts to annotate a ground-truth label for each image, our annotation approach significantly alleviates the burden on experts while yielding commendable results. Thus, we argue that this application is indeed compelling. MIPL holds considerable promise across diverse computer vision scenarios, including histopathological image classification, fine-grained image classification, video classification, and video anomaly detection. At present, MIPL remains in its nascent stage, and CRC-MIPL is one real-world dataset that will be made publicly available. Moving forward, we intend to curate additional datasets tailored to various scenarios to provide further support to the research community.
>
> References:
>
> [a] Humayun Irshad, Eun-Yeong Oh, Daniel Schmolze, Liza M Quintana, Laura Collins, Rulla M Tamimi, and Andrew H Beck. Crowdsourcing scoring of immunohistochemistry images: Evaluating performance of the crowd and an automated computational method. Scientific Reports, 7(1):43286:1-10, 2017.
>
> [b] Anne Grote, Nadine S. Schaadt, Germain Forestier, Cédric Wemmert, and Friedrich Feuerhake. Crowdsourcing of histological image labeling and object delineation by medical students. IEEE Transactions Medical Imaging, 38(5):1284–1294, 2019.
>
> [30] Wei Tang, Weijia Zhang, and Min-Ling Zhang. Multi-instance partial-label learning: Towards exploiting dual inexact supervision. Science China Information Sciences, pages 1–16, 2023.
>
> [44] Jakob Nikolas Kather, Johannes Krisam, Pornpimol Charoentong, Tom Luedde, Esther Herpel, Cleo-Aron Weis, Timo Gaiser, Alexander Marx, Nektarios A Valous, Dyke Ferber, et al. Predicting survival from colorectal cancer histology slides using deep learning: A retrospective multicenter study. PLoS Medicine, 16(1):e1002730:1–22, 2019.

---

> > ### Comment · Reviewer_pq8x · 2023-08-17
> >
> > I thank the authors for their response. The rebuttal does not significantly address my concerns. While the assumption of a single label may have been made in prior work, that does not make it realistic. My concern with the histology task was with the task and setting. It is hard to believe such a workflow could work in practice given privacy concerns for medical data. For most medical diagnosis tasks it takes significant training and years of experience to understand how to label something. It is hard to believe crowdsourced workers can be trained rapidly enough to reduce cost while still satisfying the constraints of the problem. A non-medical labeling task may have been more realistic in this context.
> >
> > While there is a contribution, I am not convinced about the significance of the setting and the work. I will maintain my score.

---

> > > ### Author Response · Authors · 2023-08-17
> > >
> > > Thank you for your prompt response. The CRC-MIPL dataset comprises seven categories for colorectal cancer classification. The images in CRC-MIPL have been preprocessed by [44]. This dataset is relatively simple within the medical domain. Consequently, employing trained crowdsourced workers to annotate this dataset can lead to satisfactory results. In the future, we plan to relax the assumption and extend MIPL to tackle more real-world challenges.

---

### Decision · Program_Chairs · 2023-09-21

**Decision:**

Accept (poster)

**Comment:**

This paper explores multi-instance partial-label learning (MIPL), a domain where bags of instances are associated with multiple candidate labels, including one true label. The proposed DEMIPL algorithm tackles this challenge by embedding multi-instance bags into single vector representations. It leverages a disambiguation attention mechanism and a momentum-based disambiguation strategy to identify the ground-truth label. The paper introduces a real-world MIPL dataset for colorectal cancer classification, highlighting DEMIPL's superior performance compared to other MIPL and partial-label learning methods.

The reviewers recognize the significance of this innovative problem setting and the potential impact of building the real-world dataset. The design of the attention mechanism is deemed reasonable, and the experimental results show promise. Most major concerns raised during the rebuttal have been addressed. However, some reviewers still have reservations about the practicality of the problem and suggest further discussion of its limitations.